# Nevus Comedonicus Syndrome Associated with Psychiatric Disorder

**DOI:** 10.3390/diagnostics12020383

**Published:** 2022-02-02

**Authors:** Ha Young Woo, Sang Kyum Kim

**Affiliations:** 1Department of Pathology, National Cancer Center, Goyang 10408, Korea; why@ncc.re.kr; 2Department of Pathology, Kyung Hee University Hospital, Kyung Hee University College of Medicine, Seoul 02447, Korea; 3Department of Pathology, Yonsei University College of Medicine, Seoul 03722, Korea

**Keywords:** nevus comedonicus, nevus comedonicus syndrome, epidermal nevus syndrome, attention-deficit hyperactivity disorder

## Abstract

Nevus comedonicus (NC) is a rare hamartoma of the pilosebaceous unit origin. The association with extracutaneous abnormalities defines NC syndrome (NCS). Fewer than 50 cases of NCS have been reported in the English literature. A 31-year-old woman presented with grouped and linear comedonal papules present from birth and located on the left buttock along Blaschko’s lines. She had a history of pediatric mood disorder combined with attention-deficit hyperactivity disorder (ADHD) from 5 years of age and was recently diagnosed with sinus bradycardia. Her skin lesion was surgically removed and microscopic findings revealed the aggregation of dilated follicular infundibula filled with prominent laminated keratin plugs, a characteristic finding of NC. This is the first report presenting NCS associated with mood disorder and ADHD. Psychiatric symptoms may represent systemic manifestation of NCS.

## 1. Introduction

Nevus comedonicus (NC) is a rare hamartoma of the pilosebaceous unit origin, first described in 1895 by Kofmann [1]. NC typically presents as grouped, dilated follicular openings containing dark keratin plugs, with a linear or zosteriform distribution. NC may affect areas without hair follicles and frequently involves the face, trunk, neck and upper limbs. Three patients with buttock involvement of NC have been reported and none presented with NC syndrome (NCS) [2,3,4]. NC presents shortly after birth in approximately half of patients and patients usually develop lesions before 10 years of age [5].

NC can occur in isolation or in combination with systemic abnormalities of the central nervous system, skeletal system, teeth and eyes. Association with extracutaneous lesions defines NCS. NCS is considered part of the epidermal nevus syndrome, referring to the association of an epidermal nevus with other developmental anomalies and it most commonly involves the central nervous system, eye and musculoskeletal system. Different types of epidermal nevus syndrome have been described, each with distinct clinical features and genetic patterns. Other syndromes considered part of epidermal nevus syndrome include the following: Proteus syndrome; Schimmelpenning syndrome (nevus sebaceous syndrome); phacomatosis pigmentokeratotica, congenital hemidysplasia, ichthyosiform erythroderma and limb defects (CHILD syndrome); angora hair nevus syndrome; type 2 segmental Cowden disease; Garcia–Hafner–Happle syndrome; and Becker nevus syndrome [6].

The extracutaneous manifestations of NCS vary widely among affected organs. A recent review paper documented that the most frequently affected organ was the eye (53.2%), followed by the skeletal system (51.1%) and brain (36.2%) [5]. In most cases, the systemic abnormalities were ipsilateral to the skin lesion. In particular, ipsilateral congenital cataract has long been considered a characteristic extracutaneous manifestation of NCS [7]. Another significant feature of NCS is digit malformations.

To the best of our knowledge, this is the first reported case of NCS associated with combined mood disorder and attention-deficit hyperactivity disorder (ADHD). This case suggests that psychiatric symptoms may also be systemic manifestation of NCS.

## 2. Case Report

A 31-year-old Korean woman was referred to our hospital with grouped and linear comedonal papules on the left buttock that had been present since birth. The papules progressively increased in size and became inflamed over 30 years. She had been suffering from chronic pediatric psychiatric disorder since she was 5 years old. She had been diagnosed with borderline personality disorder, major depressive disorder, bipolar disorder and ADHD, since childhood. She was under treatment and prescribed medications by a psychiatrist. Additionally, she was recently diagnosed with sinus bradycardia by an electrocardiogram. She had two children, both of whom were diagnosed with ADHD. Her family medical history relative to her parents and siblings was not available.

On physical examination, the papules were located on the left buttock along Blaschko’s lines, with a linear distribution of approximately 25 cm long (Figure 1). The papules were pitted in black, with mild redness around them. Her general condition was good and she had no congenital anomalies. She had no neurologic, ocular, skeletal and dental abnormalities. Her laboratory results, including complete blood cell count, blood chemistry and urine analysis, were all within normal limits.

She was admitted to the plastic surgery unit and the lesion was surgically excised under general anesthesia. The excised specimen measured 30 cm × 2 cm (Figure 2). The received specimens were serially sectioned and all divided into eight blocks. Then, these sections were processed with an automatic tissue processor and embedded in paraffin blocks that were cut into 4 µm thick slices. The sections were stained with hematoxylin and eosin (H&E) and examined under light microscopy. Microscopically, the aggregation of dilated follicular infundibula filled with prominent laminated keratin plugs was observed as well as absent or rudimentary sebaceous glands. The size of the dilated follicles varied and perifollicular inflammatory cell infiltration was observed. The epidermal layer showed mild papillomatous hyperplasia with hyperkeratosis. The histological findings were consistent with the clinical diagnosis of NC (Figure 3). Combining her psychiatric symptoms and arrhythmia history, she was diagnosed with NCS.

After surgical removal of the lesion, the overall appearance was greatly improved except for a scar. At her 2-year follow-up visit, the lesion was well healed. She had no complications and no residual or newly developed lesion was found.

## 3. Discussion

NCS, as a subtype of epidermal nevus syndrome, primarily involves the neural, ocular and skeletal systems. The systemic manifestations of NCS vary widely among affected organs: (1) eyes (cataracts, strabismus, ptosis, entropion, microphthalmia, corneal erosion and nystagmus); (2) skeletal system (finger/toe anomalies, scoliosis, vertebral malformations, leg length discrepancy, short stature, pectus excavatum, arched palate, exostoses, aplasia of the femoral head, tibial dysplasia, Paget disease, hypertrophy, widely spaced nipples and tethered tongue); (3) central nervous system (intellectual impairment, electroencephalographic abnormalities, arachnoid cysts, hemiparesis, anomalies of the corpus callosum, transverse myelitis, delayed motor development, seizures, microcephaly, learning disabilities and tethered spinal cord); (4) teeth (oligodontia, hypoplasia, wide spacing and multiple caries); (5) cardiovascular system (mitral and aortic valve disease, Alagille syndrome, atrial fibrillation and carotid aneurysm); (6) others (breast carcinoma and congenital anomalies) [5]. Due to the rarity and diversity of NCS, many phenotypes of the systemic abnormalities have been reported in only single case reports and data on systemic manifestations are still accumulating.

A recent report highlighted the importance of somatic mutations in the *NEK9* gene in NC [4]. Mutations in *NEK9*, a potential regulator of follicular cell cycle, resulted in enhanced Thr210 phosphorylation, leading to a gain-of-function, and lesional tissue showed loss of markers of follicular differentiation. Recent reports have demonstrated that recessive germline *NEK9* mutations caused skeletal abnormalities, without skin manifestations. Notably, NCS features skeletal anomalies, including scoliosis, syndactyly or the absence of fingers and supranumerary digits, implying that somatic *NEK9* mutation in bone progenitors may contribute to these NCS findings [8,9].

Our case had arrhythmia as well as skin lesions, which met the diagnostic criteria for NCS. In addition, she had been suffering with complicated psychiatric disorder, including ADHD, which was inherited by her son. Therefore, our case strongly suggests that psychiatric disorder may be relevant to the systemic manifestations of NCS. ADHD is a childhood-onset condition with symptoms of inattention, impulsivity and hyperactivity. The onset of ADHD is considered to be affected by highly multifactorial and heterogeneous factors, including both genetic and non-genetic factors, such as prenatal exposure to nicotine and alcohol, prematurity and/or low birth weight [10]. In the present case, both children of the patient were also diagnosed with ADHD, suggesting that genetic factors played a significant role in the development of ADHD in both the patient and her two children. Apart from psychiatric problems, the patient had an arrhythmia, presumed to be a cardiovascular symptom of her NCS. Although we cannot conclude the exact biological correlation between the skin lesion and psychiatric disorders in this patient, the diagnosis of NCS seems to explain all of her symptoms.

## 4. Conclusions

This is the first reported case of NCS associated with combined mood disorder and ADHD. This case suggests that psychiatric symptoms may also be systemic manifestation of NCS.

## Figures and Tables

**Figure 1 diagnostics-12-00383-f001:**
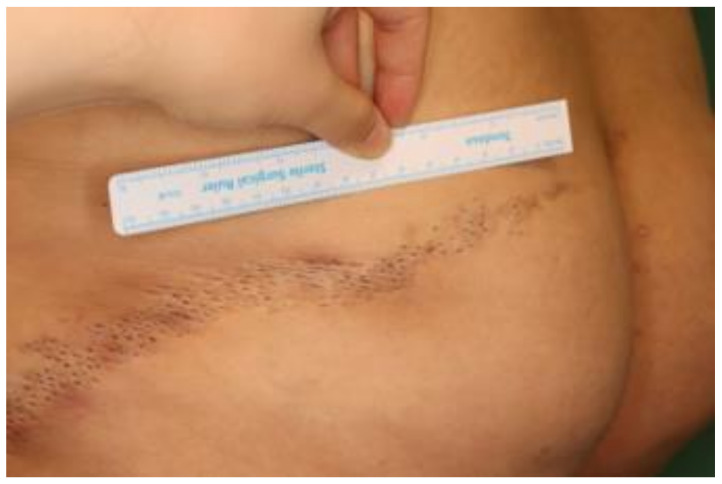
The characteristic appearance of nevus comedonicus. Grouped, linear distribution of comedo-like dark plugs on the left buttock of 31-year-old woman.

**Figure 2 diagnostics-12-00383-f002:**
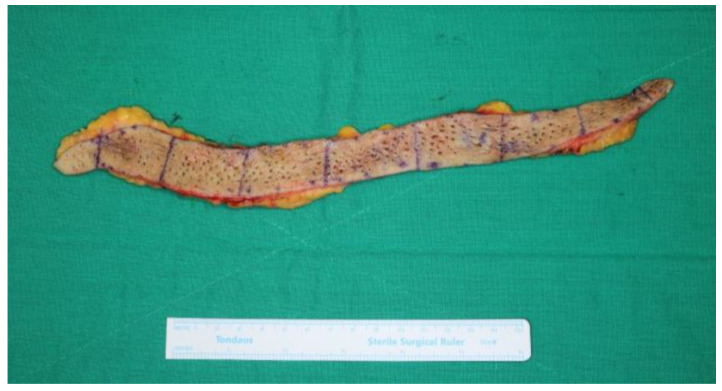
The nevus comedonicus specimen measuring 30 × 2 cm.

**Figure 3 diagnostics-12-00383-f003:**
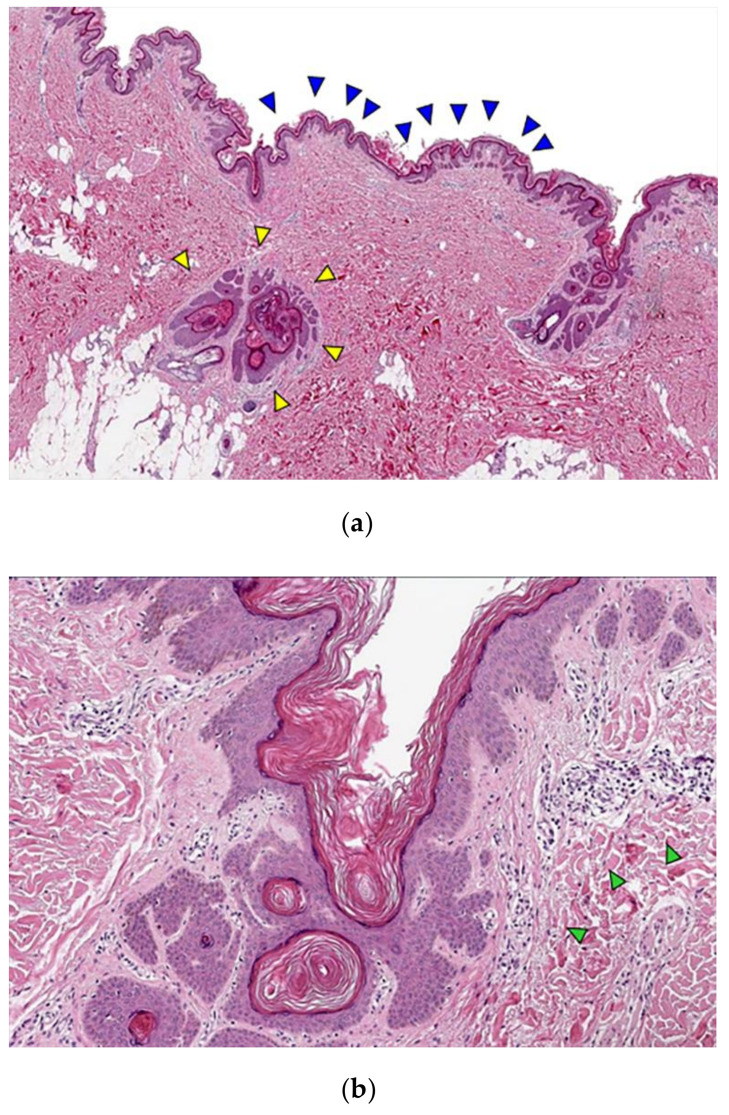
Representative microscopic images of nevus comedonicus. (**a**) Papillomatous epidermal hyperplasia (blue arrowhead) and invaginations with a laminated keratin plug (yellow arrowhead); (**b**) Mild perivascular and perifollicular lymphocytic infiltration (green arrowhead). (H & E, (**a**) ×12.5; (**b**) ×40 magnification.)

## Data Availability

The datasets generated and analyzed during the current study are available from the corresponding author upon reasonable request.

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
