# Peer review of "Nevus Comedonicus Syndrome Associated with Psychiatric Disorder"

_diagnostics, 2022, doi:10.3390/diagnostics12020383_

Round 1
Reviewer 1 Report
Fig. 3. The representative microscopic images of nevus comedonicus. (a) Acanthotic and papillomatous epidermal hyperplasia and invaginations with a laminated keratin plug; (b) Mild perifollic ular inflammatory cells infiltration. (Hematoxylin & Eosin, Original magnification; A: 1.25× objec-89 tive; B: 4× objective).
Comments:
- epidermal hyperplasia, with papillomatosis, acanthosis, invagination with the formation of multiple cystic spaces sapped by keratin, are microscopic aspects that are observed in the comedonic nevus. However, papillomatosis and acanthosis are not visible in image (a). I suggest you leave the panoramic magnification and add a higher magnification of the epidermal surface with arrows indicating papillomatosis and acanthosis respectively.
- Figure (b), on the other hand, lends itself to confusion as it shows an epithelium with so-called "proliferative" aspects that may be present in other skin lesions such as keratin cysts with proliferative aspects. I suggest you first show an overview and then the high-magnification detail of the same area; furthermore, I suggest you use the arrows to indicate the salient and specific morphological aspects for the diagnosis that you will also describe in the caption.
- I suggest you tell in the materials and methods how the sections were set up and which coloring was used, so that in the caption you can put more correctly or example like this: H&E stain x2.5, x4, x10, x40 magnification
Author Response
Reviewer 1.
Thank for your kind and insightful comments. Considering your comments, we have revised the manuscript as the follows:
Q1) Epidermal hyperplasia, with papillomatosis, acanthosis, invagination with the formation of multiple cystic spaces sapped by keratin, are microscopic aspects that are observed in the comedonic nevus. However, papillomatosis and acanthosis are not visible in image (a). I suggest you leave the panoramic magnification and add a higher magnification of the epidermal surface with arrows indicating papillomatosis and acanthosis respectively.
A1) We changed the panoramic image to a low-power one and indicated papillomatosis and acanthosis by blue and yellow arrowheads, respectively, according to the reviewer’s recommendation (Figure 3a).
Q2) Figure (b), on the other hand, lends itself to confusion as it shows an epithelium with so-called "proliferative" aspects that may be present in other skin lesions such as keratin cysts with proliferative aspects. I suggest you first show an overview and then the high-magnification detail of the same area; furthermore, I suggest you use the arrows to indicate the salient and specific morphological aspects for the diagnosis that you will also describe in the caption.
A2) Thank for your kind advice, again. We added a new image with higher magnification to show details (Figure 3b) which was magnified on the right side of Figure 3a. It shows cystic invagination, acanthosis, and keratinization. Additionally, green arrow heads show perivascular and perifollicular lymphocytic infiltrations identified in the upper dermis. These findings are included in the Figure Legend for Figure 3b as below:
“(a) Papillomatous epidermal hyperplasia (blue arrowhead) and acanthosis and invaginations with a laminated keratin plug (yellow arrowhead); (b) Mild perivascular and perifollicular lymphocytic infiltration (green arrowhead). (H&E, A: ×12.5; B: ×40 magnification).”
Q3) I suggest you tell in the materials and methods how the sections were set up and which coloring was used, so that in the caption you can put more correctly or example like this: H&E stain x2.5, x4, x10, x40 magnification
A3) As there is no ‘materials and methods’ section in this form of report, we added the detailed process of tissue management and staining in the line 74, page 2 and line 91, page 3 as below:
“The received specimens were serially sectioned and all divided into eight blocks. Then, these sections were processed with an automatic tissue processor and embedded in paraffin blocks that were cut into 4-µm-thick slices. The sections were stained with hematoxylin and eosin (H&E) and examined under light microscopy.” (line 74, page 2)
“(a) Papillomatous epidermal hyperplasia (blue arrowhead) and acanthosis and invaginations with a laminated keratin plug (yellow arrowhead); (b) Mild perivascular and peri-follicular lymphocytic infiltration (green arrowhead). (H&E, A: ×12.5; B: ×40 magnification).” (line 91, page 3)
Reviewer 2 Report
Please, don´t speak of "believe" (page 4, line 118), this is not academic, but religious language; use "in your experience" or "think".
Otherwise excellent presentation.
Author Response
Reviewer 2.
Thank for the elaborate advice, and we accepted your comment and revised the manuscript.
Q) Please, don´t speak of "believe" (page 4, line 118), this is not academic, but religious language; use "in your experience" or "think".
Otherwise excellent presentation.
A) As you pointed out, we deleted the word ‘believe’, and changed explanation as below (line 122, page 4):
“Our case had arrhythmia as well as skin lesions, which met the diagnostic criteria for NCS. In addition, she had been suffered with complicated psychiatric disorder including ADHD, which inherited to her son. Therefore, our case strongly suggests that psychiatric disorder may be relevant to the systemic manifestations of NCS.”
Reviewer 3 Report
The authors present a case report of a patient with Nevus comedonicus syndrome, that presented with psychiatric disorder, namely mood disorder and ADHD. Nevus comedonicus syndrome is a rare syndrome with variability of morphological and physiological presentations. On the other hand psychiatric disorders are common in population, some are genetically determined.
How can authors be convinced that Nevus comedonicus syndrome is associated with those psychiatric disorders? Can this be just a co-incidence?
Figure 3b has a cutting artefact.
Author Response
Reviewer 3.
Thank for the insightful comment, and we revised the manuscript according to your point.
Q1) The authors present a case report of a patient with Nevus comedonicus syndrome, that presented with psychiatric disorder, namely mood disorder and ADHD. Nevus comedonicus syndrome is a rare syndrome with variability of morphological and physiological presentations. On the other hand psychiatric disorders are common in population, some are genetically determined.
How can authors be convinced that Nevus comedonicus syndrome is associated with those psychiatric disorders? Can this be just a co-incidence?
A1) Our case had already been confirmed as NCS, as the patient had arrhythmia as well as skin lesions. In addition, she suffered from psychiatric problems, and there appears to be a genetic predisposition, especially given that ADHD was passed on to her son. Therefore, our case strongly suggests that psychiatric disorder may be relevant to the systemic manifestations of NCS. We added some explanations and tried to strengthen the association (line 122, page 4).
Q2) Figure 3b has a cutting artefact.
A2) Thank for the careful comment. We replaced it to a new picture without artifacts to improve Figure 3.
Round 2
Reviewer 1 Report
Dear Authors,
I have seen the changes. You still need to correct one small thing.
Figure 3 line 92 ".... and acanthosis and invaginations with a laminated keratin plug (yellow arrowhead);"
The laminated keratin plug is evident, while acanthosis is not seen well. I suggest either to eliminate the term acanthosis or to show a high magnification of the figure (a) (for example x40) of the epidermal line with an arrow indicating the spinous layer, hyperplasia and acanthosis. Finally, you need to correct "(H&E, A: × 12.5; B: × 40 magnification)" with (H&E, a × 12.5; b: × 40 magnification).
Best regards
Author Response
Dear reviewer,
As you pointed out, we deleted the term ‘acanthosis’ in the main text (line 81, page 3) and the figure legend (line 91, page 3). Additionally, we corrected the figure panels (line 94, page 3) as below:
“The epidermal layer showed mild papillomatous hyperplasia with hyperkeratosis.” (line 81, page 3)
“The representative microscopic images of nevus comedonicus. (a) Papillomatous epidermal hyperplasia (blue arrowhead) and invaginations with a laminated keratin plug (yellow arrowhead); (b) Mild perivascular and perifollicular lymphocytic infiltration (green arrowhead). (H&E, a: ×12.5; b: ×40 magnification).” (line 91-94, page 3)
Reviewer 3 Report
The authors have followed my comments and corrected the manuscript.
Author Response
Thank you for your kind comment.
Round 3
Reviewer 1 Report
No additional comments